# Metabolic Effects of Anti-TNF-α Treatment in Rheumatoid Arthritis

**DOI:** 10.3390/diseases11040164

**Published:** 2023-11-09

**Authors:** Kristína Macáková, Mária Tekeľová, Vanda Mlynáriková, Katarína Šebeková, Barbora Vlková, Peter Celec, Emöke Šteňová

**Affiliations:** 1Institute of Molecular Biomedicine, Faculty of Medicine, Comenius University, 81108 Bratislava, Slovakia; kristina.macakova@imbm.sk (K.M.); ma.tekelova@gmail.com (M.T.); katarina.sebekova@imbm.sk (K.Š.); barbora.vlkova@imbm.sk (B.V.); 2National Institute of Rheumatic Diseases, 92101 Piešťany, Slovakia; vanda.mlynarikova@nurch.sk; 3Institute of Pathophysiology, Faculty of Medicine, Comenius University, 81108 Bratislava, Slovakia; 41st Department of Internal Medicine, Faculty of Medicine, University Hospital, Comenius University, Mickiewiczova 13, 82101 Bratislava, Slovakia; e.stenova@hotmail.com

**Keywords:** biological therapy, metabolism, inflammation, metabolic syndrome, metabesity

## Abstract

Rheumatoid arthritis (RA) is associated with high cardiovascular mortality. It is not clear whether the metabolic consequences of chronic inflammation are involved. Biological disease-modifying anti-rheumatic drugs (bDMARDs) are highly efficient in the treatment of inflammation in RA. In this study, we aimed to describe the metabolic effects of anti-TNF-α treatment in RA patients. The clinical status of 16 patients was assessed using disease activity score-28 (DAS28) and C-reactive protein (CRP). Plasma samples were collected before treatment with anti-TNF-α treatment as well as after three and six months of treatment. Markers of lipid and glucose metabolism, as well as renal biomarkers, were assessed using standard biochemistry. ELISA was used for the quantification of insulin, leptin, and adiponectin. Although fasting insulin decreased by 14% at the end of the study, most of the analyzed parameters did not show any statistically or clinically significant dynamics. The exception was total bilirubin and cholesterol, which increased by 53% and 14%, respectively, after six months of treatment with anti-TNF-α treatment. Anti-TNF-α treatment did not induce major metabolic changes despite the strong anti-inflammatory and clinical symptoms of RA. Further studies will show whether longer observations are required for the detection of the metabolic effects of the anti-inflammatory treatment. Additional research is needed to understand the observed effect of bilirubin as an important endogenous antioxidant.

## 1. Introduction

Rheumatoid arthritis (RA) is a chronic, autoimmune, inflammatory disease identified by extensive synovitis and extra-articular manifestation leading to systemic disorders [1]. The prevalence of RA globally is around 1%, and the prevalence in Europe is between 0.2–0.4% [2]. Patients with RA have an increased risk of the development of various comorbidities; the risk of the development of cardiovascular disorders is two-fold higher than in the general population. This can be the consequence of persisting chronic inflammation and its effect on metabolism [3]. Various factors related to chronic inflammation can influence metabolism imbalance, including lipid, glucose, and hormonal metabolism [4,5]. Chronic inflammation, thus, can interfere with the pathways leading to oxidative and metabolic alterations, resulting in the increased risk of CV disease [6]. The presence of inflammation leads to changes in the properties of low-density lipoprotein (LDL) and high-density lipoprotein (HDL) [7,8]. Activation of the pro-inflammatory receptors in the immune cells can lead to fatty liver disease through the accumulation of cholesterol and other lipids. The excess of saturated fats in contact with adipocytes or hepatocytes can extend the inflammation by intensifying the leukocyte infiltration, developing the metabolic inflammatory cycle. Disturbances of the metabolic markers during RA are also involved in the lipid paradox leading to a decreased total cholesterol [9]. Insulin resistance (IR) in RA patients can be explained by the hypothesis of the selfish immune system. The immune cells require enormous energy for their activity, which is only available if other systems do not use its main source—glucose—leading to IR [10]. The metabolism is regulated by numerous hormones. Leptin and adiponectin have also been found to be important in RA [11]. Adiponectin has been shown to have pro-inflammatory properties in RA in contrast to other inflammatory diseases [12]. Leptin increases during inflammation and seems to stimulate both innate and acquired immune responses [13]. Except for the involvement in the inflammatory processes, these hormones also modulate glucose metabolism and appetite, secondarily affecting the immune response [14]. RA is treated with very efficient biological disease-modifying anti-rheumatic drugs (bDMARDs) [1]. bDMARDs target cytokines such as tumor necrosis factor (TNF) α or interleukin 6 [15]. The treatment inhibits crucial components of the inflammation reaction, resulting in the improvement of the manifestation of the ongoing inflammatory processes. bDMARDs are administered to RA patients if the traditional disease-modifying treatment with methotrexate fails [1]. Despite the powerful effects of bDMARDs, they still represent an unspecific approach for the treatment of RA, as the exact etiology is still unclear [1]. Ambiguous properties of a biological treatment can lead to various off-target responses or side effects of bDMARDs [16]. These include adverse effects on lipid or glucose metabolism [16,17]. Anti-cytokine treatment aims to decrease inflammation but also interacts with the metabolism of HDL and LDL cholesterol after treatment initiation [7,16]. The processes listed above provide a clarified insight into what might be behind the metabolic changes in RA patients that may ultimately lead to cardiovascular complications, as the stage of dyslipidemia is linked to metabolic syndrome [18]. Although a description of the mechanisms behind cranking processes may provide compelling explanations, the question remains: how do we prevent these processes? One possibility is to follow the effect of biological treatments known to be highly effective. It is known from publications that the administration of drugs in RA patients, such as methotrexate, has had beneficial effects and caused a decrease in the risk of metabolic disorders [19]. A new type of RA treatment that includes anti-TNF therapy has been shown to reduce the risk of endothelial dysfunction [20]. However, the application of anti-inflammatory therapy has not only been described at the level of proper endothelial function, but also at different levels of metabolic processes such as glucose metabolism or lipid profile [21,22,23]. The results of several studies report the effect of RA anti-inflammatory therapy as unclear and encourage further research. The administration of anti-TNF therapy not only influences metabolic changes but also changes of the oxidative stress and antioxidant status. We have described how the application of the biological treatment led to a decrease in thiobarbituric acid-reacting substances as a marker of lipid peroxidation [6]. We have also observed a decrease in extracellular DNA, which can cause a decrease in inflammation or be its consequence [24,25,26]. Given that RA also causes cartilage and bone damage that ultimately has a major effect on quality of life, the administration of anti-TNF also leads to functional benefits such as cartilage remodeling and periodontal status, as RA and periodontitis share similarities in the pathomechanism of the disease, like the production of interleukins [27,28].

Since the metabolic effects of bDMARDs treatment of RA are not clear, we aimed our study to describe the effects of anti-TNF α on metabolic outcomes in patients with RA. Thus, the aim of the work was to identify different profile groups, such as liver markers, markers describing lipid and glucose metabolism, and key hormones involved in the pathomechanisms of RA as well as in inflammatory processes.

## 2. Materials and Methods

### 2.1. Patients

In this study, starting in 2019, we collected plasma from 16 patients diagnosed with RA according to EULAR criteria [23]. The total number of patients included in the study was 77. Of those patients, 61 were excluded from the study. Exclusion criteria included major comorbidities such as cancer or heart failure, as well as treatment with other biologics in the past. The last patient was excluded because of his male sex so that the cohort is homogenous (Appendix A). The final group of patients, after the application of all exclusion and inclusion criteria, consisted of 16 women diagnosed with RA. Disease activity score-28 (DAS28) and C-reactive protein (CRP), body mass index (BMI), and erythrocyte sedimentation rate, assessed using the Wintrobe’s method [29], were used for clinical scoring assessment. DAS28, a marker evaluating pain and response to the treatment of patients, was established based on the characterization and count of 28 swollen and tender joints [30,31]. CRP was quantified using routine biochemistry (Biolis 24i Premium Instrument, Tokyo, Boeki, Japan).

Patients were treated at the rheumatology outpatient clinic of the 1st Department of Internal Medicine of University Hospital in Bratislava, Slovakia. Patients were recruited in the phase of high disease activity (disease activity score of 28 joints DAS28 > 5.1) and treated with conventional synthetic disease-modifying drugs (csDMARD) and corticosteroids. Patients started therapy with monoclonal antibodies adalimumab (n = 11, 40 mg s.c. every other week), golimumab (n = 1, 50 mg s.c. once a month), certolizumab (n = 2, 200 mg s.c. every other week), or with a circulating receptor fusion protein etanercept (n = 2, 50 mg s.c. once a week). The baseline characteristics of the patients are summarized in Table 1.

### 2.2. Blood Sampling

In this study, sampling time points were as follows: before treatment with anti-TNF-α and after three and six months of treatment. Anti-TNF-α treatment was administered continuously throughout the observed period. For the blood collection, heparin tubes (BD Vacutainer System, Plymouth, UK) were used. Plasma samples after centrifugation were frozen to −20 °C within an hour. Samples were stored at −80 °C for up to one year.

### 2.3. Metabolic Markers

Different metabolic markers were assessed in the collected blood samples. For glucose metabolism, fasting glucose and insulin concentrations were measured. From the fasting glucose and insulin, the quantitative insulin sensitivity check index (QUICKI) was calculated [32]. To assess liver function, the enzymatic activities of aspartate aminotransferase (AST) and alanine transaminase (ALT) were measured. The lipid profile included total cholesterol, high-density lipoprotein (HDL), low-density lipoprotein (LDL) cholesterol, and triglycerides (TAG). Metabolic markers, including bilirubin, were determined using routine laboratory methods (Biolis 24i Premium instrument, Tokyo, Boeki, Japan). Insulin concentration was measured using an ELISA assay (R&D systems, Minneapolis, MN, USA). Concentrations of adiponectin and leptin were also quantified with corresponding ELISA assays (R&D systems, Minneapolis, MN, USA). The relative change was calculated as the percentage of the value at the baseline point (before administration of anti-TNF-α treatment) for every individual patient. The average percentual change is reported in the comparison of the particular time point to the baseline.

### 2.4. Statistical Analysis

Statistical analysis was performed using GraphPad Prism 8.1 software (La Jolla, San Diego, CA, USA). To test the normality, the Shapiro-Wilk test was used. Parametric repeated measures One-way ANOVA was used with a Bonferroni-corrected post-hoc t-test to assess differences between time points. Results with *p*-values lower than 0.05 were considered statistically significant. Data are presented as mean + standard deviation (SD). The F-value, named after Ronald Fisher, is calculated in ANOVA and describes the ratio of the variation between the time points/variation within the time points. The t-value originally from the Student’s *t*-test measures the size of the difference relative to the variation within the time points.

## 3. Results

The efficiency of the treatment was evaluated using CRP and DAS28. CRP showed a significant decrease by 60% after 3 months and by 76% after 6 months of treatment. Between 3 and 6 months of the treatment, CRP decreased by only 16% (Figure 1: after 3 months: F = 15.73, t = 3.72, *p* < 0.01; after 6 months: F = 15.73, t = 4.4, *p* < 0.01; and between 3 months and 6 months: F = 15.73, t = 2.13, *p* > 0.05). The erythrocyte sedimentation rate showed a significant decline by 43% and 40% after 3 and 6 months, respectively (Figure 1: after 3 months: F = 8.58, t = 3.93, *p* < 0.01; after 6 months: F = 10.6, t = 2.97, *p* < 0.05). Clinical score DAS28 also showed significant changes after the application of anti-TNF α treatment. DAS28 decreased significantly by 52% and 51%, 3 and 6 months after treatment start, respectively (Figure 1: after 3 months: F = 37.21 t = 12.86, *p* < 0.001; after 6 months: F = 37.2, t = 6.27, *p* < 0.001). The BMI of patients did not change during the observation period (Figure 1: F = 0.74, *p* > 0.05). No significant effect was observed with QUICKI (Figure 2). Anti-TNF α treatment increased bilirubin significantly after 6 months by 53% (Figure 3: F = 5.1, t = 2.87, *p* < 0.05). The concentration of total cholesterol increased by 11% after 6 months (Figure 4: F = 3.29, *p* > 0.05). No significant changes in fasting insulin were induced by anti-TNF α treatment. Similarly, no significant effect was observed in the concentration of leptin, adiponectin, and their ratio (Figure 5). Similarly, no significant changes were observed within the renal biomarkers (Figure 6).

## 4. Discussion

Our results show that the anti-TNF-α treatment in patients with RA caused a major decrease in inflammatory parameters and clinical outcomes after just three months of initiation. The high efficacy of the administered therapy led to a major reduction of clinical markers describing the condition of involved patients, such as CRP, DAS28, and sedimentation rate. Results, thus, confirmed the clinical benefits of the biological therapy in RA [1,15,22]. Surprisingly, biological treatment did not induce any major changes in the observed metabolic parameters. The fact that a significant decrease in inflammation does not affect metabolic parameters is unexpected. As the etiology of RA is still unknown, even this very efficient treatment is still unspecific [33,34]. Chronic inflammation is associated with cardiovascular disease through the shared mediators between the inflammatory factors present in the synovium and those involved in atherosclerotic plaque formation [35,36]. The mechanisms of bDMARDs include the inhibition of the pro-inflammatory cytokines, leading to a decrease in inflammation. TNF α inhibitors can, thus, not only reduce clinical markers of inflammation but also reduce endothelial dysfunction and oxidative stress and modify the lipid profile or prothrombic markers [37,38]. The long-term application of bDMARDs can lead to adverse effects in the form of liver toxicity; however, every type of the bDMARDs has different tolerability. For example, infliximab can lead through an immune-mediated process to type I autoimmune hepatitis [39]. In the literature, anti-inflammatory treatments affected the lipid profile, although the induced changes in HDL and LDL are inconsistent [8,16,22]. In our study, the lipid profile was not affected by bDMARDs. These negative results are in line with a similar study describing the effects of the anti-TNF treatment in patients with RA and ankylosing spondylitis; however, there are also studies that describe the decline of LDL concentrations after the treatment application [40,41]. Taking into account the lipid paradox theory in which lower concentrations of HDL and LDL are associated with a higher CV risk [42], our results do not prove that anti-inflammatory treatment decreases the CV risk, although the dynamics of the lipid profile is compatible with previously published studies [43,44]. According to the literature, the concentration of bilirubin in RA patients was described as lower compared to healthy individuals [45]. The concentration of bilirubin was described as inversely proportional to the disease activity [46]. Thus, the increased concentration of bilirubin within the physiological range after an efficient treatment start might indicate the beneficial effect of the treatment that might be supported by the antioxidant effects of bilirubin [46,47]. Based on the literature, we expected an effect on liver enzymes [48]. However, our results do not show any effect of the bDMARDs treatment on AST or ALT, despite the fact that the literature describes that the application of anti-TNF can result in changes in liver enzymes [49]. Another factor associated with the CV risk is glucose and insulin metabolism [43,44]. Our results show no significant decrease in fasting insulin and, thus, also in QUICKI. These results contrast with studies in which the application of bDMARDs leads to decreased risk for the development of diabetes mellitus, especially in RA patients [21,50]. The possible explanation for why the markers of glucose and insulin metabolism remained without changes is due to the physiological state that was maintained during the whole observation period. Based on the literature, adiponectin and leptin could be the link between inflammation and metabolism [51]. Both hormones participate in energy and lipoprotein homeostasis, and they both affect glucose and insulin regulation [52,53]. Adiponectin was shown to positively correlate with RA duration and progression [51]. The ratio of the hormones did not change during the observation period. Comparable results were reported in a study focused on the role of adipokines in RA [8]. Unchanged concentrations of leptin and adiponectin remains unchanged even after two years of observation [54]. One of the limitations of this study is that participating patients had a healthy glucose or lipid metabolism even before bDMARDs treatment initiation, at least based on the parameters measured. A larger cohort with a wider palette of comorbidities would be more informative and could potentially show a positive metabolic effect of the anti-inflammatory treatment. On the other hand, two contradictory effects could be working against each other—improvement of the metabolism due to lowering inflammation and changes in the diet due to improved symptoms. This, however, was not monitored and can be, thus, only a matter of speculation. Of course, another major limitation of this study is the relatively short period of the observation with unknown long-term metabolic effects of bDMARDs treatment [55,56]. Additional limitations are the lack of assessment of dietary habits as they can have a direct impact on metabolism. It has been shown that changes in diet, complemented with probiotics, induce changes in TAG, HDL, and LDL [57]. In our study, we did not observe any systematic changes in the above-mentioned markers, suggesting that diet was not a major source of bias. To describe possible changes in the metabolic stage of RA patients, blood pressure and central obesity screening would be needed. Additionally, none of the patients was completely naïve to treatment, only to bDMARDs. Previous treatments could, thus, bias the observations as the disease pathogenesis and symptomatology might have started years in the past [58].

## 5. Conclusions

To conclude, this study shows that despite the major decrease in clinical markers CRP and DAS28, which confirmed the benefits of the treatment, our observations did not reveal any significant, clinically meaningful changes in the metabolism of glucose or the lipid, liver, and hormone profile.

## Figures and Tables

**Figure 1 diseases-11-00164-f001:**
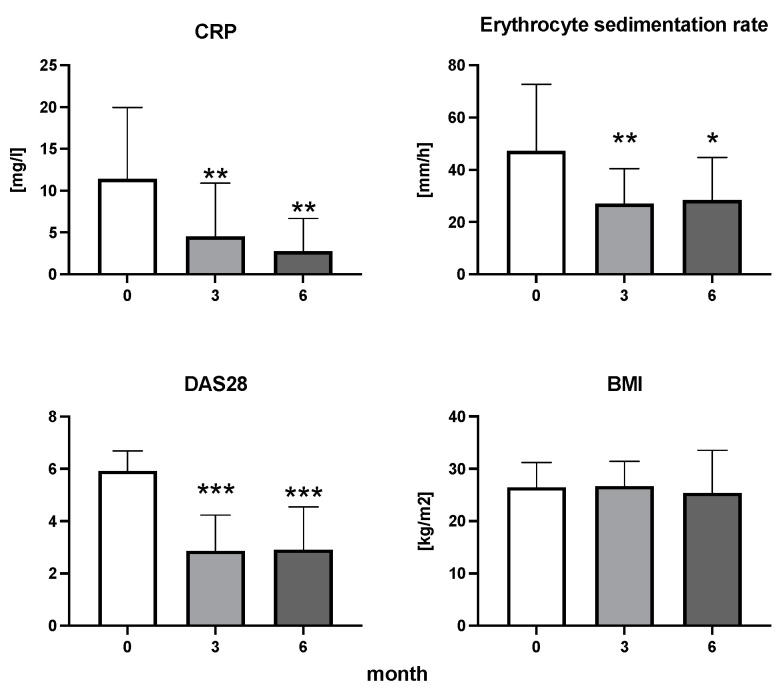
The dynamics of CRP, erythrocyte sedimentation rate, DAS28, and BMI in patients with rheumatoid arthritis before the initiation of bDMARDs and at 3 and 6 months during ongoing treatment (*p* < 0.05 = *, *p* < 0.01 = **, *p* < 0.001 = ***).

**Figure 2 diseases-11-00164-f002:**
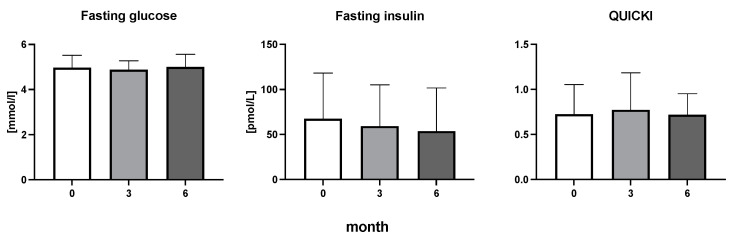
Fasting glucose, fasting insulin, and quantitative insulin sensitivity check index in patients with rheumatoid arthritis before the initiation of bDMARDs and at 3 and 6 months during ongoing treatment Table 1.

**Figure 3 diseases-11-00164-f003:**
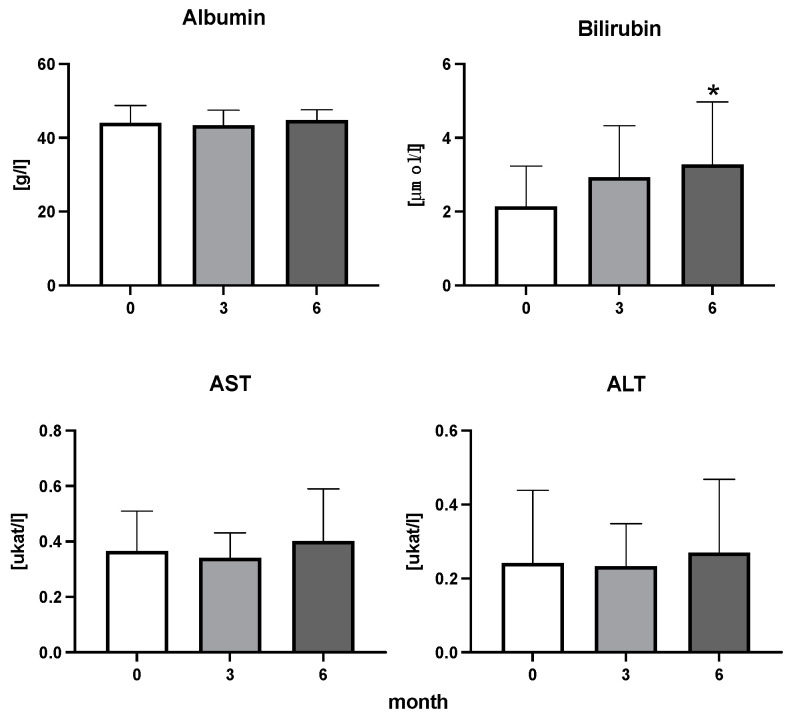
Albumin, bilirubin, enzymatic activity of aspartate aminotransferase (AST), and alanine transaminase (ALT) in patients with rheumatoid arthritis before the initiation of bDMARDs and at 3 and 6 months during ongoing treatment (*p* < 0.05 = *).

**Figure 4 diseases-11-00164-f004:**
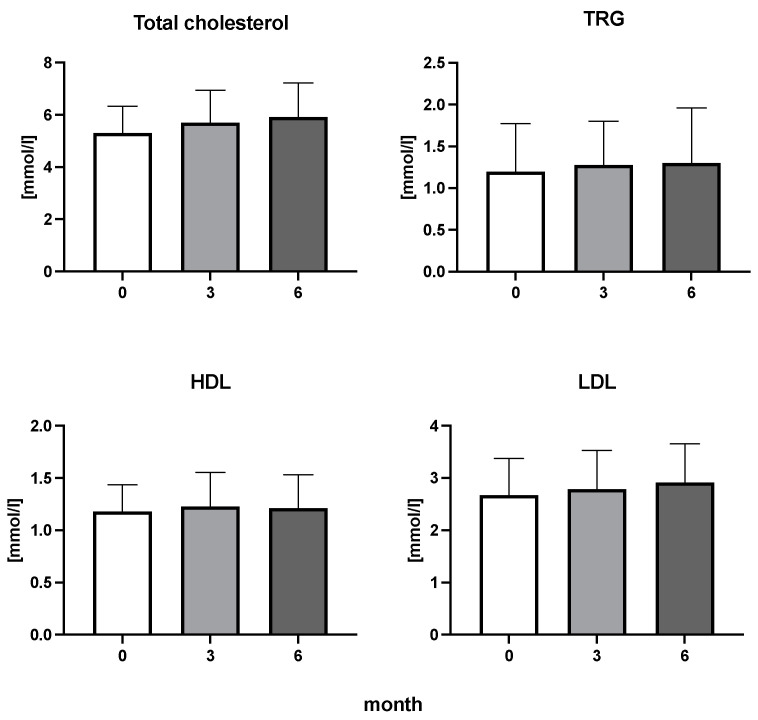
Total cholesterol, triglycerides (TRG), high-density lipoproteins (HDL), and low-density lipoproteins (LDL) cholesterol in patients with rheumatoid arthritis before the initiation of bDMARDs and at 3 and 6 months during ongoing treatment.

**Figure 5 diseases-11-00164-f005:**
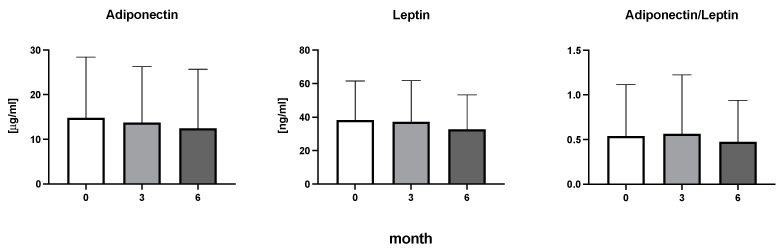
Adiponectin, leptin, and the ratio of adiponectin to leptin in patients with rheumatoid arthritis before the initiation of bDMARDs and at 3 and 6 months during ongoing treatment.

**Figure 6 diseases-11-00164-f006:**
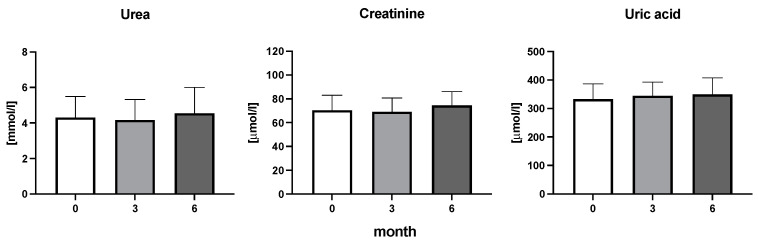
Urea, creatine, and uric acid concentration in patients with rheumatoid arthritis before the initiation of bDMARDs and at 3 and 6 months during ongoing treatment.

**Table 1 diseases-11-00164-t001:** Baseline characteristics of patients and their treatment.

Patients	Total (n = 16)
Sex female, (n)	(16)
Age, mean (±SD)	58.3 (±8.3)
Smoking history (past and current) % (n)	23.5 (4)
BMI, kg/m^2^, mean (±SD)	26.4 (±4.8)
RF + ve, % (n)	82.3 (14)
ACPA + ve, % (n)	76.4 (13)
DAS28, mean (±SD)	5.9 (±0.8)
Duration of RA (years) (±SD)	8.9 (±8.0)
Treatment	
methotrexate % (n)	76.4 (13)
leflunomide % (n)	5.9 (1)
hydroxychloroquine % (n)	5.9 (1)
sulfasalazine % (n)	17.6 (3)
cyclosporin % (n)	5.9 (1)
prednisolone % (n)	64.7 (11)

## Data Availability

The datasets generated in the current study are available from the corresponding author by request.

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
