# Peer review of "Metabolic Effects of Anti-TNF-α Treatment in Rheumatoid Arthritis"

_diseases, 2023, doi:10.3390/diseases11040164_

Round 1
Reviewer 1 Report
Comments and Suggestions for Authors
The article authored by Mackova et al., titled "Metabolic Consequences of Anti-TNF-α Therapy in Rheumatoid Arthritis," demonstrates well-structured and thoroughly documented research. Nevertheless, several concerns have arisen
The sample size utilized in the study appears to be quite limited. It is advisable for the authors to consider expanding the sample size if feasible.
The prevalence of rheumatoid arthritis within the studied populations remains unspecified. It would be beneficial for the authors to address this aspect within the main text to provide context.
Information regarding the number of patients excluded from the study and the total number of subjects recruited is not currently available. The authors should supply these crucial details for a more comprehensive understanding of their research.
Author Response
Reviewer 1
The article authored by Mackova et al., titled "Metabolic Consequences of Anti-TNF-α Therapy in Rheumatoid Arthritis," demonstrates well-structured and thoroughly documented research. Nevertheless, several concerns have arisen.
Our response: We thank the Reviewer for the positive evaluation of the topic and design.
Reviewer 1
The sample size utilized in the study appears to be quite limited. It is advisable for the authors to consider expanding the sample size if feasible.
Our response: We thank the Reviewer for the suggestion. We agree that the sample size is the major limitation of the study. We admit this in the manuscript. On the other side, this study is not a cross-sectional study, but rather a cohort study. Markers of metabolism were measured in three timepoints (before treatment, three and six months after treatment start). Samples were collected consistently from the same patients at selected time points. Total number of the recruited patients was 77. The first losses/dropouts of patients occurred not only based on exclusion criteria but also due to their absence at medical check-ups, which made it impossible to follow the dynamic in time, so that patients had to be excluded from the final evaluation. The study has already been completed and a further addition of patients would take years. Of course, we will plan more patients in our future studies. However, the patients within the group are well characterized and relatively homogeneous. Especially, after Reviewer 2 suggested the exclusion of one man participating in this study. Changed results are highlighted in the text of the revised manuscript.
Reviewer 1
The prevalence of rheumatoid arthritis within the studied populations remains unspecified. It would be beneficial for the authors to address this aspect within the main text to provide context.
Our response: We thank the Reviewer for the comment. We agree that this information was not mentioned in the manuscript. The true prevalence of RA in Slovakia is not assessed and we are not aware of any study specifically studying this issue. However, we expect that there is no major difference to other developed countries. Based on the publication from Nature reviews rheumatology by Axel Finkch in 2022 the prevalence of RA varies from 0.20-0.40% in eastern and western Europe. The prevalence of rheumatoid arthritis in Slovakia is not properly documented.
“The prevalence of RA in the world is around 1%, the prevalence in Europe is between 0.2-0.4% [2].”
Reviewer 1
Information regarding the number of patients excluded from the study and the total number of subjects recruited is not currently available. The authors should supply these crucial details for a more comprehensive understanding of their research.
Our response: We thank the Reviewer for the comment. We added the missing information about the total number of patients involved in the study as well as the number of excluded patients from the study. Total number of the recruited patients was 77, with a drop out due to missing visits and sampling and exclusion of men during the revision. Of the patients who attended regular check-ups, those patients who did not pass the study criteria were later excluded.
“In this study, starting in 2019, we used plasma from 16 patients diagnosed with RA according to EULAR criteria [23]. Total number of patients included in the study was 77, 61 patients were excluded from the study.”
Reviewer 2 Report
Comments and Suggestions for Authors
In patients with autoimmune diseases, including rheumatoid arthritis, it is important to monitor not only the classic parameters of the disease and inflammation, but also metabolic ones. Chronic inflammation is known to affect the metabolism, which can lead to complications in the future, such as cardiovascular disease. In the manuscript “Metabolic effects of anti-TNF-α treatment in rheumatoid arthritis,” the authors attempted to evaluate the effect of this type of therapy on various metabolic parameters in the blood of patients with RA. Many parameters were assessed, which is a plus of this manuscript. However, there are a number of problems due to which I believe that the manuscript in its present form cannot be published in Diseases and should be rejected. More detailed comments on the manuscript are given below.
Major Concerns
1) The sample of patients with RA is too small. 17 people are not enough to conduct a serious study with a claim to publication in Diseases. And if we consider that the study has been conducted since 2019, then it is even more difficult to accept such a number of patients.
2) No healthy control. It would be nice to have healthy controls of the same demographics at point zero to assess the levels of markers in patients' blood at the time of blood collection. And also in order to evaluate the impact of therapy, how much it brings patients closer to the indicators of healthy controls.
3) Table 1: Patient demographics, medications, etc., and tables describing their data should be presented in the «Results» section rather than in the «Materials and Methods» section.
4) As you write in the paragraph about the limitations of the study, you do not know about the diet that the patients followed during the study. It's good that you admit this, but diet can have too much influence when we talk about indices and markers like BMI, HDL, LDL and so on.
Minor Concerns
Line 32: Enter the acronym RA after the first mention of rheumatoid arthritis in the text.
Table 1: You write that there were 94.12% women in the study, but it is indicated in brackets that there was only one woman. There should probably be a number 16 here. And why is this man needed in the study at all? If you have a sample of 16 women and 1 man, it would be acceptable to remove him from the study so that the sample is less heterogeneous.
Line 118: In this line you spell out the acronyms HDL and LDL. These abbreviations appear for the first time in the line 41. Spell out them in this line. And in the line 118, leave either the full form or abbreviations.
Figure 1: How does the CRP level change at the 6-month point compared to the 3-month point? Other graphs show that all indicators that changed in the first three months go to a plateau and do not change in any way by the sixth month of the study. However, CRP levels continue to decline. Can you please write somewhere how statistically significantly it changes by the sixth month compared to the third month?
Author Response
Reviewer 2
In patients with autoimmune diseases, including rheumatoid arthritis, it is important to monitor not only the classic parameters of the disease and inflammation but also metabolic ones. Chronic inflammation is known to affect the metabolism, which can lead to complications in the future, such as cardiovascular disease. In the manuscript “Metabolic effects of anti-TNF-α treatment in rheumatoid arthritis,” the authors attempted to evaluate the effect of this type of therapy on various metabolic parameters in the blood of patients with RA. Many parameters were assessed, which is a plus of this manuscript. However, there are a number of problems due to which I believe that the manuscript in its present form cannot be published in Diseases and should be rejected. More detailed comments on the manuscript are given below.
Our response: We thank the Reviewer for the positive evaluation of the study design and the measurements of the metabolic parameters.
Reviewer 2
The sample of patients with RA is too small. 17 people are not enough to conduct a serious study with a claim to publication in Diseases. And if we consider that the study has been conducted since 2019, then it is even more difficult to accept such a number of patients.
Our response: We thank the Reviewer for the comment. The total number of the recruited patients was 77 patients. However, dropouts due to missing visits and samplings reduced the analyzed number of patients, indeed. Patients were also excluded due to exclusion criteria to obtain a more homogenous or at least less heterogenous population. This study is closed. Unfortunately, it is not possible for us to recruit more patients within this study. However, we plan a new study focusing on the new drugs and their effect. Of course, we will learn and try to get more patients or reduce the dropouts.
Reviewer 2
No healthy control. It would be nice to have healthy controls of the same demographics at point zero to assess the levels of markers in patients' blood at the time of blood collection. And also in order to evaluate the impact of therapy, how much it brings patients closer to the indicators of healthy controls.
Our response: We thank the Reviewer for the suggestion. The aim of this study was to observe/monitor the effect of the anti-TNFalpha treatment. This is only possible on patients. We, of course, agree that heatlhy controls would be also valuable. In an animal experiment this would be a must. In a clinical study in cooperation with the rheumatology department, this is simply not possible. However, as we have three samples from every included patient, the patients themselves and their baseline values represent the controls. In addition, the analyzed metabolic markers were within the physiological range.
Reviewer 2
Table 1: Patient demographics, medications, etc., and tables describing their data should be presented in the «Results» section rather than in the «Materials and Methods» section.
Our response: We thank the Reviewer for the comment. We moved the table describing data of patients from the section Materials and Methods to the section Results.
Reviewer 2
As you write in the paragraph about the limitations of the study, you do not know about the diet that the patients followed during the study. It's good that you admit this, but diet can have too much influence when we talk about indices and markers like BMI, HDL, LDL and so on.
Our response: We thank the Reviewer for the comment. In our study we found no major change in the metabolic parameters. We are aware that diet affects metabolism. But that would be a problem especially if we had observed changes that could be explained by the diet or the drugs. No change means that likely neither the treatment nor the diet had a major effect. This, however, does not mean that the diet should not be monitored. We have added the information about the potential effect of diet on the metabolic markers in our study in the discussion.
“Additional limitations are the lack of assessment of the dietary habits as they can have a direct impact on the metabolism. It has been shown that the changes in the diet complemented with probiotics induce changes in TAG, HDL and LDL [57]. In our study we did not observe any systematic changes in the above mentioned markers suggesting that the diet was not a major source of bias.”
Reviewer 2
Line 32: Enter the acronym RA after the first mention of rheumatoid arthritis in the text.
Our response: We thank the Reviewer for the suggestion. We added the acronym RA into the manuscript where it was mentioned for the first time.
Abstract: “Rheumatoid arthritis (RA) is associated with high cardiovascular mortality. It is not clear whether the metabolic consequences of chronic inflammation are involved.” Introduction: “Rheumatoid arthritis (RA) is a chronic, autoimmune, inflammatory disease identified by extensive synovitis and extra-articular manifestation leading to systemic disorders [1].”
Reviewer 2
Table 1: You write that there were 94.12% women in the study, but it is indicated in brackets that there was only one woman. There should probably be a number 16 here. And why is this man needed in the study at all? If you have a sample of 16 women and 1 man, it would be acceptable to remove him from the study so that the sample is less heterogeneous.
Our response: We thank the Reviewer for the suggestion. We removed the one man from the study to make the cohort more homogenous. We performed a new statical analysis, we prepared new graphs and changed the information about the patients involved in the study in the section Material and Methods,
“In this study, starting in 2019, we used plasma from 16 patients diagnosed with RA according to EULAR criteria [23]. Total number of patients included in the study was 77, 61 patients were excluded from the study.”
Reviewer 2
Line 118: In this line you spell out the acronyms HDL and LDL. These abbreviations appear for the first time in the line 41. Spell out them in this line. And in the line 118, leave either the full form or abbreviations.
Our response: We thank the Reviewer for the comments. We spelled the acronyms of HDL and LDL in the line 41. We left the full name in the line 118.
“The presence of inflammation leads to changes in the properties of low-density lipoprotein (LDL) and high-density lipoprotein (HDL) [7, 8]. Activation of the pro-inflammatory”.
“To assess liver function enzymatic activities of aspartate aminotransferase (AST) and alanine transaminase (ALT) were measured. The lipid profile included total cholesterol, high-density lipoprotein (HDL), low-density lipoprotein (LDL) cholesterol, and triglycerides (TAG).”
Reviewer 2
Figure 1: How does the CRP level change at the 6-month point compared to the 3-month point? Other graphs show that all indicators that changed in the first three months go to a plateau and do not change in any way by the sixth month of the study. However, CRP levels continue to decline. Can you please write somewhere how statistically significantly it changes by the sixth month compared to the third month?
Our response: We thank the Reviewer for the suggestion. The CRP concentrations between the 3 months and 6 months after the treatment initiation changed by 14%. We added the information about the details of the statical changes of CRP in time, between third and sixth month of the treatment, into the section Results.
“CRP showed a significant decrease by 60% after 3 months and by 76 % after 6 months of treatment. Between the 3 months and 6 months of the treatment, CRP decreased only by the 16% (Figure 1, after 3 months: F=15.73, t(15)=3.72, p<0.01, after 6 months: F=15.73, t(15)=4.4, p<0.01, between 3 months and 6 months F=15.73, t(15)=2.13, p>0.05).”
Reviewer 3 Report
Comments and Suggestions for Authors
This manuscript mainly described the metabolic effects of anti-TNF-α treatment in RA patients. After reading the manuscript carefully, I can tell you that the information it brings to readers is reasonable and interesting. However, In my opinion, there are several minor revisions for this manuscript. 1.English, grammar and typos need to be further improved. 2.In Abstract, the full name of DAS28 and CRP should be added. 3.In Introduction section, more information should be added to describe the treatment effect of anti-TNF-α on RA and their own research results should also be cited. 4.The authors should add some related research results of their own research group, not just other people’s literatures. 5.Please add some lastest references in the manuscript, especially references in recent two years. 6.In the Discussion section, the authors should compare their own research results with previous reports, especially the different results need to be further discussed.
Comments on the Quality of English LanguageEnglish, grammar and typos need to be further improved.
Author Response
Reviewer 3
This manuscript mainly described the metabolic effects of anti-TNF-α treatment in RA patients. After reading the manuscript carefully, I can tell you that the information it brings to readers is reasonable and interesting. However, In my opinion, there are several minor revisions for this manuscript.
Our response: We thank the Reviewer for the positive evaluation of the topic and the importance of the results.
Reviewer 3
1.English, grammar and typos need to be further improved.
Our response: We thank the Reviewer for the comment. The revised manuscript has been checked by a native speaker.
Reviewer 3
2.In Abstract, the full name of DAS28 and CRP should be added.
Our response: We thank the Reviewer for the comments. We added full name of DAS28 and CRP into the abstract.
“The clinical status of 16 patients was assessed using disease activity score - 28 (DAS28) and C-reactive protein (CRP).”
Reviewer 3
3.In Introduction section, more information should be added to describe the treatment effect of anti-TNF-α on RA and their own research results should also be cited.
Our response: We thank the Reviewer for the suggestion. We have added more information about the effect of the treatment from our own research into the Introduction.
“Administration of anti-TNF therapy not only influences metabolic changes but also changes of the oxidative stress and antioxidant status. We have described how the application of the biological treatment led to a decrease in thiobarbituric acid-reacting substances as a marker of lipid peroxidation 6. We have also observed a decrease in extracellular DNA, which can cause a decrease in inflammation or be its consequence [24-26]. Given that RA also causes cartilage and bone damage that ultimately has a major effect on quality of life, administration of anti-TNF also leads to functional benefits such as cartilage remodeling and periodontal status, as RA and periodontitis shared similarities in the pathomechanism of the disease, like productions of interleukins [27-28].”
Reviewer 3
4.The authors should add some related research results of their own research group, not just other people’s literatures.
Our response: We thank the Reviewer for the suggestion. We added the related research results also from our own research group.
“Administration of anti-TNF therapy not only influences metabolic changes but also changes of the oxidative stress and antioxidant status. We have described how the application of the biological treatment led to a decrease in thiobarbituric acid-reacting substances as a marker of lipid peroxidation 6. We have also observed a decrease in extracellular DNA, which can cause a decrease in inflammation or be its consequence [24-26]. Given that RA also causes cartilage and bone damage that ultimately has a major effect on quality of life, administration of anti-TNF also leads to functional benefits such as cartilage remodeling and periodontal status, as RA and periodontitis shared similarities in the pathomechanism of the disease, like productions of interleukins [27-28].”
Reviewer 3
5.Please add some lastest references in the manuscript, especially references in recent two years.
Our response: We thank the Reviewer for the suggestion. We have added references no older than two years. New citations are highlighted in the revised manuscript.
Reviewer 3
6.In the Discussion section, the authors should compare their own research results with previous reports, especially the different results need to be further discussed.
Our response: We thank the Reviewer for the suggestion. In the discussion we discussed the current results from the study with our previous results from our own research. Changes are highlighted in the discussion.
Round 2
Reviewer 2 Report
Comments and Suggestions for Authors
I accept all your responses to my comments in the previous round of review. I see no further obstacles to the publication of your manuscript. However, please describe in more detail why you excluded so many people from the study. Please, make a flow-chart diagram for this. Here is an example of such a diagram, Figure 2. https://journals.plos.org/plosone/article?id=10.1371/journal.pone.0203492 (this article does not need to be cited, this is just an example). A detailed description of this chart can be found here (https://stats.stackexchange.com/questions/457885/what-is-the-name-of-a-chart-that-visualizes-the-inclusion-and-exclusion-of-patie). You can choose where to add this diagram, in the main text of the manuscript or in the supplementary material.
Author Response
Response to reviewers
We would like to thank the Reviewer 2 for the constructive suggestion. We hope that the revised version of the manuscript is now suitable for publication in Diseases.
Reviewer 2
I accept all your responses to my comments in the previous round of review. I see no further obstacles to the publication of your manuscript. However, please describe in more detail why you excluded so many people from the study. Please, make a flow-chart diagram for this. Here is an example of such a diagram, Figure 2. https://journals.plos.org/plosone/article?id=10.1371/journal.pone.0203492 (this article does not need to be cited, this is just an example). A detailed description of this chart can be found here (https://stats.stackexchange.com/questions/457885/what-is-the-name-of-a-chart-that-visualizes-the-inclusion-and-exclusion-of-patie). You can choose where to add this diagram, in the main text of the manuscript or in the supplementary material.
Our response: We thank the Reviewer for the acceptance of our responses. Based on the example we prepared a flow-chart diagram to describe the exclusion of patients from the study. We added the flow-chart diagram into the supplementary material of the manuscript.